# The assignment and distribution of the dyslexia label: Using the UK Millennium Cohort Study to investigate the socio-demographic predictors of the dyslexia label in England and Wales

**Cathryn Knight** *, **Tom Crick**

School of Social Sciences, Swansea University, Swansea, United Kingdom

* cathryn.knight@swansea.ac.uk

## Abstract

The prevalence of dyslexia identification has increased significantly over the last two decades. Yet there is debate over whether there are distinct biological and cognitive differences between those with literacy difficulties and the subgroup of people identified as dyslexic. This is the first paper that provides evidence for this ongoing debate by investigating the socio-demographic factors, outside biology and cognition, that predict whether a child is identified as dyslexic in the UK. Using secondary data from the UK's Millennium Cohort Study, this paper examines the socio-demographic factors that predict whether a child's teacher identifies them as dyslexic at age 11. Gender, season of birth, socio-economic class and parental income are found to be significant predictors of the dyslexia label. Therefore, factors seemingly unrelated to the clinical aspects of dyslexia influence whether a child is identified as dyslexic in England and Wales. This suggests that label may not be evenly distributed across a population; furthermore, it may also indicate that resources for support may not be fairly allocated. The findings further support the argument that a 'dyslexic subgroup' within poor readers is created due to the impact of environmental factors. The results from this national-scale study thus questions the reliability, validity and moral integrity of the allocation of the dyslexia label across current education systems in the UK.

## Introduction

Dyslexia is commonly understood as a problem with decoding the written word which impacts a person's ability in literacy-based activities. In 2012–2013, the number of students entering higher education institutions in the UK who had been identified as dyslexic was 22 times higher than the number entering two decades previously (1994–1995) [1]. Yet, recent debates question the reliability and validity of the label. Whether there is any difference between those who have received a diagnosis of dyslexia and those who struggle with literacy has been systematically questioned. Research in the field suggests that there is no compelling evidence that

**Data Availability Statement:** The data that support the findings of this study are openly available in UK data service at: http://doi.org/10.5255/UKDA-SN-5350-5, reference number [5350-5]; http://doi.org/10.5255/UKDA-SN-5795-5, reference number

[5795-5]; http://doi.org/10.5255/UKDA-SN-6411-8, reference number [6411-8]; http://doi.org/10.5255/UKDA-SN-7464-5, reference number [7464-5]; http://doi.org/10.5255/UKDA-SN-8156-7, reference number [8156-7].

**Funding:** CK ES/J500197/1 Economic and Social Research Council (ESRC) https://esrc.ukri.org/ The funders had no role in study design, data collection and analysis, decision to publish, or preparation of the manuscript.

**Competing interests:** The authors have declared that no competing interests exist.

the etiology, neurology, neuropsychology, or treatments for 'dyslexia' and 'low reading achievement' are different [2–10]. Therefore, to identify a subgroup of people with dyslexia may be misleading. Due to the lack of evidence of clear biological or cognitive differences between those with literacy difficulties and those who are identified as dyslexic, this paper investigates what other variables lead to a person being assigned the dyslexia label, by their teacher, outside of their literacy difficulties.

While previous research has investigated socio-demographic predictors of special educational needs (SEN) in England [11], this is the first paper to investigate the dyslexia label specifically. This is of particular interest as, while there is cognitive and biological evidence for other subgroups of SEN, clear evidence of these differences in those with dyslexia is lacking [2, 5]. Given that dyslexia identification can lead to resources and support such as extra time in examinations and additional educational input [12], it is vital to know whether this resource allocation is equitable in its distribution among the population. This paper uses secondary data from the UK's Millennium Cohort Study (MCS) to investigate how the dyslexia label is assigned and distributed across the population, and thus, what socio-demographic factors predict who is identified in a dyslexic subgroup at the end of primary school.

## The dyslexia label

The advantages and disadvantages of labelling with a special educational need (SEN) have been long debated [13–16]. With regards to dyslexia, Riddick [17] states that while the identification of dyslexia may be beneficial to the individual, it is not beneficial at the cohort or population level as it marks the individual as being different from others. Elliott [18] takes a wider look at the dyslexia labelling process and suggests that dyslexia identification—and subsequent resourcing—might benefit those who are labelled; however, those who do not receive the label, but who also may be struggling with reading, thus do not receive the support they may need.

Qualitative research tends to point towards a positive impact of labelling with SEN due to the alleviation of stigma and giving the child an explanation for why they may have been struggling academically [19–21]. On the other hand, Knight [22] demonstrates that those who are labelled as dyslexic by their teacher or parent generally have lower academic outlook and aspirations in comparison to matched peers, highlighting a potential negative impact of the dyslexia label. Missing from these debates, however, is a discussion of who is being labelled with dyslexia in the first place.

## Socio-demographic predictors of the dyslexia label

While little research has been conducted looking specifically at the socio-demographic predictors of dyslexia, King and Bearman [23] investigated the likelihood of a child having a diagnosis of autism in the USA. They aimed to investigate 'the way in which individual and neighbourhood characteristics interact over time to shape health outcomes' (p.321). Their results showed that along with individual-level factors, community-level variables increased the likelihood of a child being diagnosed with autism. 'Children born to wealthier and more educated parents, living in wealthy neighbourhoods have the highest probability of obtaining an autism diagnosis' [23]. Therefore, they argue that an understanding of the complex cultural and environmental aspects surrounding those with autism, is as important as understanding the underlying biological aspects of the condition. This paper aims to develop a better understanding of the impact of these individual and cultural factors on those who have been labelled with dyslexia. Given the complex nature of dyslexia identification, it is not possible, or plausible, to identify every factor that may be associated with the dyslexia label. Furthermore, the motivation for this paper is also driven by concerns about who is receiving the label of dyslexia,

be this via an official diagnosis or more informal processes within the education system. The following discussion identifies socio-demographic factors that the literature suggests may be associated with the dyslexia label.

**Gender.**   There is evidence to suggest that fewer girls are diagnosed with dyslexia than boys. It has been found that in most school systems, girls tend to be better at reading [24, 25]; whilst there is also evidence that more boys than girls have the phonological awareness difficulties commonly associated with dyslexia [26]. Miles, Haslum and Wheeler [27] present evidence which shows that more males have reading disabilities. Furthermore, results from the Organisation for Economic Co-operation and Development's (OECD) Programme for International Student Assessment (PISA) [28] shows sex differences in reading skills across countries, suggesting that sex differences in reading are not down to educational practices or language orthography and that they are likely biological in nature. Further research has shown that sex differences in reading skill are more apparent at the lower end of the reading distribution [29, 30], suggesting that biological sex may be involved in reading difficulties such as dyslexia.

However, in studies of those that have been referred for a dyslexia assessment, the male: female ratio ranges from 3:1 to 5:1, whereas, in epidemiological studies of dyslexia, a smaller disparity has been found (1.5:1 to 3.3:1) [31]. This suggests an over-referral of males for dyslexia assessments compared to females. Shaywitz [32] spoke of the 'myth' of male vulnerability to reading disability (p.98). Shaywitz suggests that the reason more boys than girls receive a diagnosis of dyslexia is that girls are often less obtrusive and attention-seeking [32]. Therefore, boys are brought to the attention of teachers more easily, causing a disproportionate referral of boys. It could be argued that these characteristics may interplay with biological academic ability in leading to dyslexia identification.

This raises interesting questions about the biological differences found between boys and girls, and how this may interact with environmental norms with regards to displays of gender. What is unclear here is whether there are biological differences which explain why more boys than girls have been found to have dyslexia, or whether there are environmental aspects at play which influence the perception of difficulties in boys and girls differently. Arnett et al. [29] suggests that the overrepresentation of males can be due to two factors 'one invalid part explained by referral bias, and one potentially valid residual part found in epidemiological samples' (p.719). To follow this up they studied sex differences in a sample of 2,399 7- to 24-year-olds and found a small male:female disparity (1.15:1). They concluded that 'the higher prevalence of males with reading difficulties can be explained by a combination of males' slower and more variable processing speed and worse inhibitory control, although these are partly offset by males' better verbal reasoning' (p.726). These findings suggest that there are valid sex differences between males and females; however, the extent of these differences is not as large as the referral rates suggest. Therefore, there appears to be a complex system in the referral, diagnosis and labelling of dyslexia. While there may be some evidence for a biological underpinning in the differences between males and females, this does not entirely explain the large discrepancy in referrals for diagnosis.

**Age in year group.**   Donfrancesco et al. [33] found that in Italy, those who were younger in the year when starting school were more likely to be identified as dyslexic. However, the relationship between age in year group and the dyslexia label remains unexplored in the UK.

The impact of age in year group has become a growing area of interest; Crawford, Dearden and Greaves [34] produced a report on the effect of being born in the spring or summer and therefore being young in the academic year in England. They found 'large differences in educational attainment between children born at the start and end of the academic year in England' [34]. These differences decreased as the child got older, but they state that the gap

remains significant throughout compulsory schooling. Large-scale longitudinal datasets are the best ways to follow these trends as they can assess the impact of the month of birth over an individual's life course. Research using the MCS found that older children in the year group were significantly more likely to be placed in the highest set, whereas those who were younger were more likely to be in the lowest set [35].

Relationships have also been found between a child's birthday and SEN. Crawford, Dearden and Greaves [34] show that 'relative to those born in September, those born in August are 5.4% more likely to be labelled as having mild special educational needs at age 11' (p.2). Furthermore, Zoega, Valdimarsdóttir and Hernández-Díaz [36] researched the relationship between attention deficit hyperactivity disorder (ADHD) and month of birth. They found that children in the youngest third of a class were 50% more likely than those in the oldest third to be prescribed stimulants between ages 7 and 14. They conclude that the effect of age should be considered in the diagnostic process of ADHD.

Therefore, it is of interest to investigate whether children in England and Wales with the dyslexia label are more likely to be younger in their year group. As there is unlikely to be a biological reason why someone who is younger in their year group would be more likely to have dyslexia, should this be the case, it will highlight the importance of social processes in who is labelled with dyslexia.

**Ethnicity.**   Evidence from the USA shows ethnic disproportionality in SEN labelling [37], whereby those from minority ethnic groups are more likely to be labelled with a SEN. Yet Morgan et al. [38] found that when controlling for confounding variables, including individual child-level academic achievement and behavioural functioning as well as family-level socioeconomic status, minority children are less likely than otherwise similar White, English-speaking children to be identified as disabled and receive special education services in the USA. Strand and Lindsay [39] conducted a study of SEN and ethnicity in England using school census data and found that while being a member of a minority ethnicity was a predictor of SEN, gender and poverty were stronger predictors. They also found differences between SEN types and minority ethnic groups. While they did not look specifically at dyslexia, analysis of those with a specific learning difficulty (SpLD) showed that when controlling for gender and poverty, those from *Mixed White and Asian*, *Indian*, *Pakistani*, *Bangladeshi*, *Other Asian*, and *Chinese* ethnic groups were significantly less likely to have a SpLD than *White* students [39]. In addition, Parsons and Platt [11] found that white children in the MCS were more likely to be considered dyslexic when controlling for indicators of deprivation. This, therefore, suggests an overrepresentation of white people being considered as dyslexic.

**Social class.**   A highly debated topic is the impact of social class on the identification of dyslexia. Media reporting in the UK has often promoted a notion that dyslexia is a 'middle class' phenomenon, present in fee-paying schools in particular [40, 41]. Yet, a 2010 policy paper on special education by the UK government noted the strong correlation between having SEN and being from a disadvantaged background [42]. However, within the wider SEN category, dyslexia was not found to be correlated with higher levels of disadvantage [11]. Tomlinson [43] states that

a relatively undocumented theme is that much of the expansion of special education categories and demands for funding and resources have, from the 1980s, come from middle class and articulate parents who [. . .] have seized on expanding ways of defending children in need of special education and support

(p. 273–274).

Therefore, it is relevant to consider the impact of social class on dyslexia identification, and in particular, the impact of belonging to the middle class.

Reay, Crozier and James [44] claim that 'above all, the distinguishing feature of the middle classes is a particular set of values, commitments and moral stances [. . .] such as ambition, sense of entitlement, educational excellence, confidence, competitiveness, hard work and deferred gratification' (p.12). These values suggest that great emphasis is placed on education in middle class children. In an interview survey of working class and middle class parents, Reay [45] found that parents' involvement in schooling was largely class-based; she argued that the middle classes are able to utilise their economic and cultural resources to ensure that there is a continuation of their children's educational advantage.

Gillborn [46] researched the experiences of black middle class parents with children who had SEN. He found that within his participants it was usually the parent who identified the problem and sought assessment for the child. He stated that 'this involves drawing on both their economic capital (the financially expensive specialist assessments) and their cultural and social capital (often using friendship and professional networks to help 'negotiate the system') (p.280); this discussion of capital builds on the work of Bourdieu [47]. In brief, Bourdieu suggested that a key divide between the middle and working class was the volume and composition of habitus possessed by those in the different social classes. Bourdieu proposed that habitus was made up of the type and amount of capital that a person has. Capital can be made up from economic capital (financial resources), social capital (the individual's social network) and, cultural capital (the knowledge of the right cultural codes of how to behave in various social contexts). This habitus then affects how the individual interacts with the field (in this case education). Reay [45] describes how middle-class parents have cultural capital in a combination of 'requisite skills and competencies, confidence in relation to the educational system, a previous history of being supported educationally in the home, educational knowledge and information about schooling' (p.111). Relating to dyslexia, findings from Gilborn's [46] study suggest that parents in the middle class are able to mobilise their economic resources in order to obtain a diagnosis for their child. Furthermore, they are able to use their cultural capital in the form of social networks and knowledge of the field to get the most out of the education system for their children.

Having a label of dyslexia can lead to adjustments in formal examinations [12]. As qualifications 'are designed and delivered to differentiate pupils and students' [48, p.107] and middle class parents have been shown to be motivated to ensure that their child exceeds in education [44, 49, 50] it is easy to see why gaining adjustments in examinations may be important to middle class parents. In contrast, working class parents may not have the economic resources to pay for an assessment. Furthermore, they may not have access to the networks and educational history that firstly, may inform them about the potential benefits of dyslexia identification, and secondly, may help them in obtaining this label.

Furthermore, other factors that may also be associated with the construct of social class. Parsons and Platt [11] and Anders et al. [51] show correlation between parents having lower educational levels and their child having a SEN, however this has been unexplored with dyslexia specifically. Furthermore, according to the British Dyslexia Association (BDA) the cost of assessment 'is £540 (+ VAT) with a specialist teacher and £720 (+ VAT) with an Educational Psychologist' [52] (prices correct as of July 2021). Given the cost of dyslexia diagnosis, the impact of parents' income also needs to be taken into consideration.

It is of interest to understand how the norms and values of different social classes may impact whether or not a child is labelled with dyslexia and furthermore, how the current dyslexia system may be contributing to the class structures that we see in the current education system.

### The present study

This study is interested in how the aforementioned socio-demographic variables may influence whether or not someone is identified as dyslexic at the end of primary school (age ≈ 11). Should the identification of dyslexia be due to biological and cognitive factors alone, then there would be little evidence to suggest that it is in some way nested in particular demographics. Being labelled as dyslexic may be seen as desirable to some as it may allow for access to more educational resources and interventions, such as extra time in exams; as well as potentially reducing the 'embarrassment' of struggling with literacy [17]. Thus, should environmental factors be involved in dyslexia diagnosis, then particular individuals may be more likely to have dyslexia than others. Should this be the case, knowledge of these predictors will help to ensure that that all those that are showing the same difficulties are receiving the appropriate help, intervention and resources.

Therefore, the research aims to answer the following overarching research question:

- What socio-demographic factors (i.e. gender, age in year group, ethnicity, and social class) influence the probability of dyslexia being identified by a child's teacher at the age of 11?

## Methods

### Data

Data from the UK's Millennium Cohort Study (MCS) was used [53–57] for this study. The MCS is a large-scale longitudinal study which aims to study a sample cohort of approximately 19,000 babies born between 1 September 2000 and 31 August 2001 in England and Wales, and between 24 November 2000 and 11 January 2002 in Scotland and Northern Ireland. The MCS covers a diverse range of topics; to date there have been seven data sweeps. The current paper will focus on outcomes from sweep five (published in 2012) when the children were approximately 11 years old and at the end of primary school. While the MCS now provides data from families in England, Wales, Scotland and Northern Ireland, this analysis uses data from the Teacher Survey at age 11 that was only conducted in England and Wales; therefore, only data from England and Wales was analysed. Whilst education in each UK nation is a devolved policy area, at the time of data collection for sweep five (2011) both counties had similar policies and resource allocation for special educational needs. Furthermore, the country was controlled for during analysis to account for any country-specific differences. The MCS provides weights for each sweep to account for attrition and oversampling of particular sub-categories; the country-specific weight was applied for the analysis.

### Variables

**Dyslexia.** At age 11, teachers were asked to comment on whether the child has dyslexia. If the teacher responded 'yes' to '*Does this child have Special Educational Needs (SEN)*?', they were asked to select the reason from a list which included dyslexia. The teachers identified 253 children as dyslexic.

While this variable allowed insight into whether or not the teacher believed that the child had dyslexia, it did not allow insight into whether or not the child had officially been diagnosed. Previous research has used administrative data on the National Pupil Database (NPD) in England linked with the MCS in order to establish whether the child had an officially recognised SEN [58]. However, the NPD does not provide information on the type of SEN. Rather, the NPD dataset provides and overarching category of SEN, broken down into the type of support that the cohort member has received. As this research was interested in the dyslexia label

**Table 1. Those labelled as dyslexic and on the SEN register at age 11, KS2.**

|  | No SEN (KS2) | SEN (KS2) | Total |
|---|---|---|---|
| Not dyslexic | 5,791 | 1,535 | 7,326 |
| Dyslexic | 31 | 121 | 152 |
| Total | 5,822 | 1,656 | 7,478 |

in particular, it would not have been appropriate to use this more inclusive category. Due to this, we cannot assume that the children identified as dyslexic by their teacher had been officially diagnosed as dyslexic. However, using the NPD data, it was possible to see how many of those identified as dyslexic in England, had also been officially identified with a SEN at age 11 on the NPD. Table 1 shows the number of cohort members identified as dyslexic in England at age 11, and those that are also identified on the SEN register. Of the cohort members with NPD data, 152 were identified as dyslexic at age 11. Of these 121 (79.6%) were officially identified as having a SEN on the Key Stage 2 (the legal term for the four years of schooling in maintained schools in England and Wales for pupils aged 7–11) NPD. This, therefore, suggests a fairly accurate reporting of dyslexia as a SEN by teachers at age 11.

**Age in year group.** An 'age in year group' variable was created by allocating those who would be the oldest in the year '12' (i.e. those born in September), and those youngest '1' (i.e. those born in August). However, this did not consider that parents of children who are young in their year group are able to choose to defer their school entry to the following year. Therefore, a second variable was created which provided information on how old the child was when they started school full-time. This was created using the child's month and year of birth, along with the parents' report of the month and year that the chid started school full time. The variable was created in months and ranged from children starting school at 35 months (2.9 years) to 70 months (5.8 years) with an average of 56 months (4.7 years). Using this information, those that started school younger than 4 years (47 months) and older than 5 years (60 months) were excluded from the analysis. This meant that age in year group, according to month of birth, could be examined.

**Parents' highest socio-economic class.** As part of looking at social class, the socio-economic classification (SEC) of each parent was derived from their occupation, using categories provided by the Office of National Statistics (ONS), the UK's largest independent producer of official statistics and its recognised national statistical institute. These were collapsed into five main categories: '*Managerial and professional*'; '*Intermediate*'; '*Small employer and self-employed*'; '*Low supervisors and technical*'; and, '*Semi-routine and routine*'. Using this information, the five-class structure was reverse recoded and the highest SEC for the main parent and partner parent was derived. This provided a household SEC level, using the highest SEC household member's status.

**Income.** The MCS collected information on the main and partner parents' gross earnings at each sweep. From this information, the MCS calculate the OECD-equivalised weekly family earnings. This is done by 'dividing the total net household income, with the number of household members, according to their weight on the OECD equivalised income scale (equivalised household size) to give net disposable income' [59] (p.49). The current research made use of the continuous equivalised income scale; this was used in order to determine how a one-unit change in equivalised income impacted the likelihood of the dyslexia label.

**Parents' highest education level.** The parents' highest academic or vocational qualification level was calculated. The qualifications are aggregated into a five-point scale from England and Wales' National Vocational Qualification (NVQ) level 1 (no qualifications at GCSE level)

to NVQ level 5 (higher degree and postgraduate qualification). However, due to a small number of parents of children with dyslexia holding to lowest level of education (n = 14), NVQ levels 1 and 2 were aggregated to make a four-point scale.

**Fee-paying school.** In each survey, the main parent was asked to state whether or not the child attended a fee-paying school. This response allowed the creation of the 'fee-paying school' variable.

**Control variables.** *Cognitive profile*. Cognitive profile variables were entered as control variables. Therefore, directional estimates of the likelihood of dyslexia identification, uniquely attributable to the variables of interest, were adjusted so as to make the children as otherwise similar as possible in ability. For example, it might have been found there was a relationship between dyslexia and social class, yet without controlling for ability this relationship could be spurious. For example, research has shown that children with higher non-verbal ability (i.e., an indicator of IQ) are more likely to be diagnosed with dyslexia because many clinicians and teachers still embrace an IQ-discrepancy model of dyslexia, which requires an average to above average IQ [60–63]. Higher non-verbal ability has also been found to be correlated with higher social class [64–67]. Therefore, it might appear that there is an association between higher social class and the dyslexia label, but this is in fact an artefact of a reliance on IQ-discrepancy definitions of dyslexia.

Furthermore, it is necessary to control for verbal ability (of which a low score may be directly attributable to dyslexia) so that the difference in those identified as dyslexia, and those who are not, could not be attributed to difference in verbal ability that may also cluster within the variables of interest. Therefore, controlling for verbal and non-verbal ability in the regression analyses provides a stronger test of whether social class, and other variables of interest, are independently associated with the dyslexia label than bivariate analyses alone.

These scores were derived from cohort members' age 8 results on the British Ability Scales (BAS) tests for word reading and pattern construction. These tests are widely validated, age-appropriate tests that have been shown to be predictive of later child cognitive performance [68].

*Country*. In order to control for the effect of either living in England or in Wales, the child's country at sampling and country at interview were used. Those who did not have the same country at both sampling and the time of the interview were coded as missing and were not included in the analysis. This ensured that we were able to control for the impact of consistently living in one country.

## Analysis methods

**Bivariate analysis.** Following the initial selection of variables of interest bivariate analysis was carried out using Stata. The purpose of bivariate analysis was to establish whether there were significant relationships between the aforementioned variables and the group of children labelled with dyslexia. However, bivariate analysis does not take into account any other variable which may be influencing the relationship between the two variables examined, therefore, logistic regression was also conducted.

**Logistic regression.** Logistic regression analysis allowed exploration of how the predictor variables related to the dependent variable, while taking the other variables in the model into account. As the dependent variable was binary and categorical (labelled as dyslexic/not labelled as dyslexic), binary logistic regression was used. In this study 'stepwise backwards logistic regression' was conducted. This procedure involved all possible variables being entered into the model initially and the insignificant variables being removed individually. The backwards

method was chosen as forward approaches can result in a more significant variable suppressing the effects of less significant variables so that they do not appear to be significant.

## Results

### Bivariate analysis

Table 2 shows the results of chi-square analysis exploring the relationships between being labelled with dyslexia at age 11 and the categorical socio-demographic variables. Table 3 shows the results of t-tests exploring the relationship between being the subgroup of children labelled with dyslexia and the continuous variables. These tables show that being male, attending a fee-paying school and being younger in the year group individually show a significant relationship with being labelled with dyslexia at age 11. During bivariate analysis, no relationship was found between those in the subgroup labelled with dyslexia and parents' highest socio-economic class, parents highest education level, ethnicity or income. However, as discussed above, bivariate analysis does not take into account how different variables may be influencing the relationship between significant variables. Therefore, logistic regression was used to analyse the relationships between these variables.

**Table 2. Bivariate analysis of dyslexia and categorical characteristics.**

| Group | Category | Unweighted N | Dyslexic age 11 | | Weighted % |
|---|---|---|---|---|---|
| | | | Unweighted % | Weighted N | |
| **Parents highest socio-economic class** | Semi-routine and routine | 59 | 3.4 | 66 | 3.8 |
| | Low supervisory and technical | 13 | 3 | 15 | 3.5 |
| | Small employers | 32 | 3.4 | 28 | 3.1 |
| | Intermediate | 25 | 2.5 | 28 | 2.9 |
| | Managerial and professional | 81 | 4.4 | 82 | 4.8 |
| **Ethnicity** | White | 217 | 3.8 | 230 | 3.9 |
| | Mixed | 8 | 3.3 | 9 | 3.5 |
| | Indian | *3* | *1.5* | *2* | *1.3* |
| | Pakistani or Bangladeshi | *5* | *1.2* | *3* | *1* |
| | Black or black British | 6 | 2.5 | 7 | 3.3 |
| | Other Ethnic group | 1 | 1 | 1 | 1.3 |
| **Fee paying school** | No school fees | *199* | *3.1* | *206* | *3.3* |
| | School fees | **38** | **11** | **37** | **11.1** |
| **Gender** | Male | **157** | **4.7** | **170** | **5** |
| | Female | *80* | *2.4* | *77* | *2.3* |

Those in **bold** had a z score of +1.96 meaning that this category was significantly more likely to be identified as dyslexic at age 11, those in *italics* has a z score of -1.96 meaning that this category was significantly less likely to be identified as dyslexic at age 11.

**Table 3. Bivariate analysis of dyslexia, age in year group and income.**

| | Age in year group | | | Income | | |
|---|---|---|---|---|---|---|
| | $\bar{x}^*$ | Std. Err | p | $\bar{x}$ | Std. Err | p |
| Not identified as dyslexic age 11 | 6.57 | 0.05 | 0.06 | 418 | 5.5 | 0.23 |
| Identified as dyslexic age 11 | 6.12 | 0.24 | | 432 | 12.8 | |

* 12 = Oldest, 1 = Youngest.

## Logistic regression

Following investigation of the relationship between variables in bivariate analysis consideration was given to which variables should be included in the regression models. Although not every predictor variable showed a significant relationship with the children identified with dyslexia, an initial aim was to include all predictor variables in the initial models, with insignificant variables being removed individually. This is because a bivariate analysis result may be insignificant due to 'noise' from other variables that strongly predict being labelled with dyslexia.

However, school fees were not included as bivariate analysis consistently showed a highly significant relationship between school fees and the subgroup of children labelled with dyslexia (p<0.001 in every sweep); yet the numbers of cohort members in the school fees cells were very small (Table 2). Therefore, there was a much more clustered distribution in those who attended a fee-paying school than in those who did not. As the variable was so skewed, it may supress the effects of other variables in the model; for this reason, the school fees variable was not entered into the models. However, it is important to remember its importance in its relationship with the dyslexia label.

Thus, variables that were entered into the logistic regression models were verbal and non-verbal ability (word reading and pattern construction respectively), country, gender, age in year group, parents' highest SEC, parents' highest NVQ, income and ethnicity.

Backwards stepwise logistic regression revealed parents' highest NVQ was the least significant predictor being identified as dyslexic followed by ethnicity, therefore these were removed using a stepwise method, and the model run again. This revealed that significant predictors of being identified as dyslexic were: being male, age in year group (as age in year group decreases the likelihood of being identified as dyslexic increases), having parents in the highest socio-economic class (managerial and professional), and income (as income increases the likelihood of being identified as dyslexic also increases). While country was not a significant predictor of being identified as dyslexic, it was left in in order to control for any effect of country. Table 4 shows the final regression statistics.

## Discussion

The present study aimed to answer, 'what socio-demographic factors influence the probability of dyslexia being identified by a child's teacher at the age of 11?'. Regression analysis showed

**Table 4. Logistic regression predicting the dyslexia label at age 11.**

| Variable | Category | Odds Ratio | Std. Err. | t | p | 95% Conf. Interval | |
|---|---|---|---|---|---|---|---|
| Verbal ability | | 0.92 | 0.01 | -12.35 | 0.00 | 0.91 | 0.93 |
| Non-verbal ability | | 1.00 | 0.01 | 0.51 | 0.61 | 0.99 | 1.02 |
| Country | England (ref) | | | | | | |
| | Wales | 0.55 | 0.17 | -1.96 | 0.05 | 0.30 | 1.00 |
| Age in year group | | 0.94 | 0.02 | -2.76 | 0.01 | 0.89 | 0.98 |
| Income | | 1.00 | 0.00 | 3.62 | 0.00 | 1.00 | 1.00 |
| Socio-economic class | Semi-routine and routine (ref) | | | | | | |
| | Low supervisors and technical | 0.74 | 0.35 | -0.64 | 0.52 | 0.30 | 1.86 |
| | Small employer and self-employed | 1.17 | 0.34 | 0.56 | 0.57 | 0.67 | 2.06 |
| | Intermediate | 1.17 | 0.41 | 0.44 | 0.66 | 0.59 | 2.32 |
| | Managerial and professional | 2.40 | 0.55 | 3.84 | 0.00 | 1.53 | 3.76 |
| Gender | Male (ref) | | | | | | |
| | Female | 0.57 | 0.10 | -3.07 | 0.00 | 0.40 | 0.82 |
| Constant | | 101.02 | 65.90 | 7.08 | 0.00 | 27.97 | 364.92 |

that age in year group, income, socio-economic class and gender were significant predictors of being labelled with dyslexia. We will now discuss the implications of this, before drawing concluding remarks about the dyslexia label.

## Gender

A significant individual predictor of being labelled with dyslexia was being male. Previous research suggests that due to biological and cognitive factors alone, the ratio of males to females should be 1.15:1 [29]. Results from this study suggest that, when holding the other variables in the model constant, the ratio of males to female is approximately 2:1. Therefore, there appears to be an overrepresentation of males, which cannot be explained by the biological factors suggested by Arnett et al. (2017) alone [28]. This suggests that there may be some social determinants that are also involved in why males are more likely to be labelled as dyslexic compared to females.

As previously discussed, Shaywitz [32] suggests that the reason more boys than girls receive a label of dyslexia is that girls demand less attention in the classroom and, therefore, boys are more readily brought to the attention of teachers. Furthermore, Peterson and Pennington [69] suggest that the reason for the over referral of boys is that 'boys with dyslexia come to clinical attention more often than girls, simply because they have higher rates of comorbid externalising disorders, including attention deficit hyperactivity disorder (ADHD)' (p.1997). This would suggest a biological predeterminate of the differences in diagnostic rates between boys and girls. Further research should investigate this complex interaction between the biological and social factors associated with gender and how they may result in an overrepresentation of male dyslexics.

## Age in year group

Results found that those who were younger in their year group were more likely to be labelled with dyslexia than those who were older in their year group. This result supports other research into SEN which has also found that those who are younger in their year are more likely to have a SEN [34, 36, 51]. As there is unlikely to be a neurobiological reason as to why those that are younger would be more likely to have dyslexia, this suggests that social factors are impacting the labelling process. A possible reason is that, due to being younger in the year, an individual may be underperforming in comparison to their peers resulting in a dyslexia label being sought.

## Social class structures

Parents' highest SEC was also found to be a significant predictor of being labelled with dyslexia at age 11. This effect was largely driven by the 'managerial and professional' group. Therefore, whilst studies into SEN more generally have found that it is those in the lower SECs that tend to have a SEN [11, 51, 53, 58, 70, 71], in the case of dyslexia, the opposite was found. Unfortunately, it is not possible to determine how the children studied gained the dyslexia label; however, it is plausible to suggest that those within this higher socio-economic class would have been more able to seek out and pay for a diagnosis than those in other socio-economic classes. However, it is interesting to note that both income and socio-economic class were significant predictors of the dyslexia label, showing that SEC is significant even when controlling for income. Therefore, it does not seem to be a case that being able to afford the diagnosis alone leads to a dyslexia labelling; factors associated with socio-economic class also appear to be important in the child receiving a label.

In his work with black middle-class parents, Gilborn [46] suggested that both the economic capabilities of the middle class, and their cultural and social capital aided them in getting a SEN diagnosis for their child. As SEC was found to be as important as income in predicting whether a child has dyslexia, the findings from this research suggest that the cultural and social capital of the middle class is important to take into account when considering who has been labelled with dyslexia. Therefore, it is interesting to question what social and cultural capital the highest socio-economic class has which means that they are more likely to have a child labelled with dyslexia. As discussed previously, Reay, Crozier and James (2011) suggest that a key value for the middle class is 'educational excellence' [44, p.12]. It could be hypothesised that this drive for educational excellence, combined with economic, social and cultural capital, means that the highest socio-economic class is able to manipulate their circumstances to ensure that their children get the help that they need. Furthermore, obtaining a label of dyslexia, may give the parents a reason for why their child is not showing the 'educational excellence' that they value. This raises some questions about fairness of access to the resources that are available to the subgroup of students that are identified with dyslexia.

Thus, the findings from this study not only show that social class is linked with dyslexia labelling, but can allow insight into one of the many ways that social class structures may be being reinforced in the UK. As those with the dyslexia label may have access to extra time in examinations this will arguably give them an advantage in exams in comparison to similar ability peers who do not have extra time. As economic and cultural capital appears to be a factor in dyslexia labelling, those that receive this advantage in examinations are likely to be from a more advantaged background. Consequently, providing this service to those in the highest socio-economic class, leaves those from less advantaged backgrounds without access to this additional support. Thus, this could be seen as a contributing factor in widening the education gap between the higher and lower socio-economic classes. Future research in this area should seek to explore how relevant resources for individuals with literacy difficulties varies according to these socio-demographic factors.

Furthermore, bivariate analysis showed the largely significant effect of attending a fee-paying school and being labelled with dyslexia. This, again, raises further questions about the equitability of who is able to access the dyslexia label. As the children studied are about to enter secondary school, it is particularly relevant when considering the allocation of extra time in examinations.

## Limitations

While this study has indicated a number of interesting and relevant factors that are associated with a child being labelled with dyslexia by their teacher at age 11, it is only possible to speculate as to what it is about these attributes that may lead to this identification. Therefore, further research is needed to understand exactly how these elements interact to result in dyslexia identification at age 11.

Secondly, determination of whether a child was dyslexic was made by teacher report. It is possible that a teacher's awareness of dyslexia could vary systematically as a function of the demographic variables the study is exploring. However, as shown in Table 1, 79.6% of those identified as dyslexic by their teachers were also identified as having a SEN on the KS2 NPD. This, therefore, suggests a fairly accurate reporting of dyslexia as a SEN by teachers at age 11. However, it is imported to note that the results of this study present the socio-demographic factors related to teacher identification of dyslexia, and not necessarily dyslexia diagnosis by an educational psychologist or similar.

## Concluding remarks

Looking at dyslexia in this way further raises questions about the way that dyslexia is understood and addressed. Other researchers in the field argue that there is a lack of evidence for a 'dyslexic subgroup' who show clear differences from struggling readers [2–10]. This paper has provided evidence that factors unlikely to be related to dyslexia impact the likelihood of an 11-year old having this label. This, therefore, supports the argument that this sub-group of students labelled with dyslexia is somewhat trivially created due to the interplay of environmental and biological factors.

The findings also suggest that the dyslexia label is not evenly distributed across the population. Aspects such as month of birth were significant predictors of the dyslexia label, despite there it being unlikely that there are any biological reasons as to why this may be the case, highlighting the role that a child's environment can play in dyslexia labelling. These dimensions should be considered in the allocation of the dyslexia label. Tomlinson [43] states that 'in the current global recession governments find it easier to focus on individual deficiencies'. However, results from this analysis highlight the importance of looking further than the child to understand why someone is labelled with dyslexia. Bearman [72] questions 'what if the sequencing phenomenon is to be found not in the genome but instead in a better understanding of the social and cultural factors that shape health?' (p.11) This, therefore, suggests that while significant resources are put into understanding the genetic and biological determinates of health aspects such as dyslexia, perhaps equal attention needs to be given to the environmental factors which this paper shows are also likely to alter the probability of identification.

The findings also inform the debate about whether or not we should label with dyslexia [2–6, 12]. The results demonstrate that those with greater socio-economic and cultural capital are more likely to be labelled with dyslexia by their teachers than those without. As the label can result in additional support and extra time for assessments, it could be argued that these resources are not being fairly allocated to those who need them the most. Conversely, if the label has a negative impact on academic outlook, as found by Knight [22], this further adds to the calls from Elliott et al. [2–6, 12, 14] that the label of dyslexia should be retired. As teachers [73] have been found to hold negative ideas and perceptions about what dyslexia is, and the important role they play in shaping a young person's academic outlook, it is therefore interesting to note to whom they are assigning this label.

In 2006, the United Nations General Assembly confirmed the Convention of Rights of Persons with Disabilities [74], which included a significant commitment to inclusive education; to date (July 2021), 164 countries have signed the convention and 182 had ratified it. Many countries have shown significant steps towards inclusive education in their political rhetoric and their legislation; for example in Wales, the Additional Learning Needs Education Tribunal (Wales) Act 2018 [75] calls for a 'fully inclusive education system' [76, 77]. Yet, inclusive education is more than simply the 'integration' of students with different needs in the same class or environment [78, 79]. For a truly inclusive education system, adaptation needs to be made to teaching methods, curriculum, assessment and accessibility to ensure that all students can be meaningfully included in learning [78]. As SEN policies globally aspire to a more inclusive education system the results of this research call into question the inclusivity of how children are labelled with dyslexia. Tomlinson [43] states that:

'Inclusion should be based on a broad conception of social justice in education, moving from the endless categorisation and re-categorisation of young people judged to be failures in the system to universal learning that accommodates the diversity of all students'

(p. 275).

Yet the results from this research, along with the increasing numbers of SEN diagnosis across education systems internationally, suggest that this aim of inclusion may not yet be met (or be likely to be met) under current policy approaches. Therefore, this paper argues that changes to the system are needed to ensure equitable identification of dyslexia and subsequent support.

## Author Contributions

**Conceptualization:** Cathryn Knight.

**Formal analysis:** Cathryn Knight.

**Funding acquisition:** Cathryn Knight.

**Investigation:** Cathryn Knight.

**Methodology:** Cathryn Knight.

**Writing – original draft:** Cathryn Knight, Tom Crick.

**Writing – review & editing:** Cathryn Knight, Tom Crick.

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
