## [Decision Letter · Decision Letter 0]

9 Jun 2021

PONE-D-21-02074

The assignment and distribution of the dyslexia label: Using the UK Millennium Cohort Study to investigate the sociodemographic predictors of dyslexia in England and Wales

PLOS ONE

Dear Dr. Knight,

Thank you for submitting your manuscript to PLOS ONE. After careful consideration, we feel that it has merit but does not fully meet PLOS ONE’s publication criteria as it currently stands. Therefore, we invite you to submit a revised version of the manuscript that addresses the points raised during the review process. In particular, please expand the theoretical framework as suggested by Reviewer 2, acknowledging that there is an ongoing wide debate about the nature of dyslexia, also from a dimensional perspective.

We look forward to receiving your revised manuscript.

Kind regards,

Enrico Toffalini, Ph.D

Academic Editor

PLOS ONE

Journal Requirements:

2. Please improve statistical reporting and refer to p-values as "p<.001" instead of "p=.000". Our statistical reporting guidelines are available at https://journals.plos.org/plosone/s/submission-guidelines#loc-statistical-reporting

Reviewers' comments:

Reviewer's Responses to Questions

**Comments to the Author**

1. Is the manuscript technically sound, and do the data support the conclusions?

Reviewer #1: Yes

Reviewer #2: Yes

2. Has the statistical analysis been performed appropriately and rigorously? 

Reviewer #1: I Don't Know

Reviewer #2: Yes

3. Have the authors made all data underlying the findings in their manuscript fully available?

Reviewer #1: Yes

Reviewer #2: No

4. Is the manuscript presented in an intelligible fashion and written in standard English?

Reviewer #1: Yes

Reviewer #2: Yes

5. Review Comments to the Author

Reviewer #1: Thank you for the opportunity to review this manuscript that reported an interesting and important study. The following comments are offered to support revision of the manuscript -

The title could be further developed to emphasize that the study investigated factors correlated with the application of the diagnostic label – as opposed to ‘dyslexia’ per se. Similarly, the page 3 subheading and text with reference to ‘dyslexia’ and ‘autism’, please consider further clarifying the use of the label as opposed to specific condition (there may be other areas of the manuscript where this may be appropriate too).

I also suggest that specific / explicit statement of the research question/s are included in the 'The Present Study' subsection, and for explicit links to these in the discussion.

Thanks for a well-written manuscript.

Reviewer #2: I am not sure about the answer to question 3 above.

This is an interesting manuscript that looks at demographic factors associated with having a formal dyslexia diagnosis among 11-year-olds in the UK. I have two major questions/concerns as well as a few more minor suggestions for improvement of the work.

My primary concern begins with a question about how the authors frame the ‘debate’ that motivates the manuscript. I disagree with citing only Elliott on this question in the introduction. I think the view now widely held among experts in the field is that the terms “dyslexia,” “word-level reading difficulties,” “specific learning disability in reading,” and several others are interchangeable. In other words, dyslexia represents the low tail of the distribution of basic literacy skills (word reading, decoding, fluency, spelling). Apart from a few exclusionary conditions (e.g., literacy problems due to lack of instruction, sensory disabilities, or intellectual disability) there is not compelling evidence that the etiology, brain bases, neuropsychology, or treatments for “dyslexia” and “low reading achievement” are different. We know unquestionably that a substantial portion of children have difficulties acquiring basic literacy skills in the absence of these exclusionary criteria and that those difficulties are functionally impairing. Thus, to me, the question is not so much whether dyslexia is a valid diagnosis, but how access to accurate diagnosis and relevant resources for individuals with literacy problems varies according to the demographic factors the authors are exploring.

Related to this issue, there are multiple points throughout the manuscript where the authors refer to whether individuals “have dyslexia” where I think what they mean is whether the person has been “diagnosed with dyslexia” or “identified as having dyslexia.” (E.g., lines 28, 124, 230, 385/386, 520). This may seem like a minor point, but I think it is important to the framing of the manuscript and the take-home message. For many medical and psychological diagnoses, we know there are disparities across demographic groups in getting an accurate diagnosis as well as access to treatments. Take hypertension as an analogy. Let’s say we looked at a large group of people and found those from higher SES backgrounds were more likely to have a formal diagnosis of hypertension, but that the risk of having objectively measured high blood pressure was actually higher in individuals from a lower SES background. I don’t think we would conclude that hypertension is associated with higher SES, or that those from higher SES are more likely to have “hypertension” while those from lower SES are more likely to have “high blood pressure.” I think we would more likely say that lower SES conveys increased risk for hypertension AND that there is an SES-related disparity in how often accurate diagnosis is made. Also important, this disparity would not cause us to question whether hypertension is a valid or meaningful diagnosis.

A more minor, but related point, is that in the current manuscript, determination of whether a child had been diagnosed with dyslexia was made solely by teacher report. It is possible that teacher awareness of children’s diagnoses (or of the meaning of a dyslexia label) could vary systematically as a function of some of the demographic variables the study is exploring. The authors allude to the limitations of this approach in the methods, but I was surprised that they did not include a limitations section in the discussion reviewing this and other limitations of the work. I also suggest being clearer throughout the manuscript by using language such as “having a teacher-reported dyslexia diagnosis” or similar.

My second major concern has to do with how the authors handled IQ (the “Cognitive profile” section on page 14). I did not understand exactly what they did or why. This issue should be set up better in the introduction and probably linked to the questions that motivates the manuscript. The field has moved away from IQ-discrepancy definitions of dyslexia again because of empirical evidence that the underlying causes of, and appropriate treatments for, dyslexia do not depend on IQ score. Therefore, I think it is probably not appropriate for primary analyses to control for verbal and nonverbal abilities. However, given the historic emphasis on IQ discrepancy in defining learning disabilities, it could be appropriate to conduct secondary analyses that control for IQ and see whether the pattern of results changes. If the authors go that route, I suggest explaining the motivation for their approach (if possible, along with a priori hypotheses) in the introduction.

Minor points:

Line 69: “child having autism” – better to say “being diagnosed with autism”?

Line 185: I think there is a typo in this line

Line 252: “educational” – should be “education”?

Paragraph beginning on line 440: The Shaywitz and Peterson/Pennington explanations are different versions of the same thing, not competing explanations. One of the reasons that girls tend to demand less attention in the classroom than boys is because they have lower rates of hyperactive and impulsive symptoms.

6. PLOS authors have the option to publish the peer review history of their article (what does this mean?). If published, this will include your full peer review and any attached files.

Reviewer #1: No

Reviewer #2: No

---

## [Author Response · Author response to Decision Letter 0]

22 Jun 2021

We thank the two reviewers and editor for their useful comments and feedback on our manuscript; the resulting changes have clearly improved our paper and the clarity of its contributions, and we now feel it is ready for publication in PLOS ONE. Please see the 'response to reviewers' document attached where we clearly set out how we have addressed the reviewers' feedback.

---

## [Decision Letter · Decision Letter 1]

22 Jul 2021

PONE-D-21-02074R1

The assignment and distribution of the dyslexia label: Using the UK Millennium Cohort Study to investigate the socio-demographic predictors of the dyslexia label in England and Wales

PLOS ONE

Dear Dr. Knight,

Thank you for submitting your manuscript to PLOS ONE. After careful consideration, we feel that it has merit but does not fully meet PLOS ONE’s publication criteria as it currently stands. Therefore, we invite you to submit a revised version of the manuscript that addresses the points raised during the review process.

In particular, Reviewer 2 has raised a series of further comments, mostly about methodological clarifications, that may help improving your manuscript. Based on the Reviewer's reccomendation and my own reading, all the points raised can be considered minor, but I recommend that you carefully address all of them before acceptance.

We look forward to receiving your revised manuscript.

Kind regards,

Enrico Toffalini, Ph.D

Academic Editor

PLOS ONE

Journal Requirements:

Additional Editor Comments (if provided):

Reviewers' comments:

Reviewer's Responses to Questions

**Comments to the Author**

1. If the authors have adequately addressed your comments raised in a previous round of review and you feel that this manuscript is now acceptable for publication, you may indicate that here to bypass the “Comments to the Author” section, enter your conflict of interest statement in the “Confidential to Editor” section, and submit your "Accept" recommendation.

Reviewer #2: (No Response)

2. Is the manuscript technically sound, and do the data support the conclusions?

Reviewer #2: Yes

3. Has the statistical analysis been performed appropriately and rigorously? 

Reviewer #2: Yes

4. Have the authors made all data underlying the findings in their manuscript fully available?

Reviewer #2: Yes

5. Is the manuscript presented in an intelligible fashion and written in standard English?

Reviewer #2: Yes

6. Review Comments to the Author

Reviewer #2: Thank you for the opportunity to review this revision. The authors have addressed most of concerns while retaining the strengths of the original manuscript. I have a few remaining questions/suggestions.

Page 6, second paragraph, line 4: Do you mean “more boys than girls” instead of “more girls than boys”?

Page 8: The data on the relationship between ethnicity and special education eligibility in the U.S. are complex, and this is a controversial topic. While a higher percentage of ethnic minorities are identified for special education than white students, this is driven largely by inclusion in categories related to behavior. There is actually evidence that minority students are under-represented in special education categories most relevant to learning disabilities (including dyslexia, and so of relevance to the current paper). (Morgan et al., 2015). The authors may want to tweak this section to reflect the complexity of this issue.

Morgan, P. L., Farkas, G., Hillemeier, M. M., Mattison, R., Maczuga, S., Li, H., & Cook, M. (2015). Minorities are disproportionately underrepresented in special education: Longitudinal evidence across five disability conditions. Educational Researcher, 44(5), 278-292.

Page 15: I still find the explanation of what exactly the authors did with IQ and why confusing. Also, this issue is still not addressed in the introduction. Based on what I understand from the paper, here are my suggestions.

First, the authors actually do look at the associations between variables of interest both with and without accounting for IQ (i.e., the bivariate results on page 19 versus the regression results beginning on page 20). To the extent that the pattern of results is similar in both cases, this would strengthen the conclusions they can draw. Perhaps they can highlight this a bit more.

Second, I think the example given on page 15-16 about why it is “necessary” to control for IQ is confusing. I don’t agree that it is necessary, but I agree it provides a stronger test of some of their research questions. The explanation might be clearer if the authors gave an example of how IQ could cause a spurious relationship, rather than no relationship. I think the most likely way in which IQ would confound the results is as follows: Children with higher IQ are more likely to be diagnosed with dyslexia because many clinicians and teachers still embrace an IQ-discrepancy model of dyslexia, which essentially requires an average to above average IQ. Higher IQ could also be associated, on average, with higher social class for a wide variety of reasons. Therefore, it might appear that there is an association between higher social class and the dyslexia label, but this is really an artifact of a reliance on IQ-discrepancy definitions. Therefore, controlling for IQ in the regression analyses provides a stronger test of whether social class and other variables of interest are independently associated with the dyslexia label than the bivariate analyses.

Page 21—I recommend explaining the direction of all these effects here (e.g., which gender, which direction for age in year group, which direction for SES, education, and income) rather than in the opening of the discussion.

Page 22—As mentioned above, I think most of the first paragraph of the discussion section would fit better in results

Concluding remarks: here and in a few other places throughout the paper I think the language about factors being “unrelated to dyslexia” or there being “no biological reason” may be too strong. For example, there are a number of possible reasons that factors associated with SES could play a true etiologic role in the development of dyslexia or other learning disabilities, and the current paper cannot address that directly. Similarly, there are potential biological reasons that being born in a certain month could convey risk for particular developmental outcomes (e.g., higher likelihood of exposure to certain prenatal infections). I agree those are not the most likely explanations for the results in this paper, but again, the design cannot address them directly. So it might be better simply to soften the language throughout to say things like “unlikely to be related.” I also recommend being careful not to imply that somehow all biological effects are genetic and all environmental effects are not biological. Many environmental effects act directly on biology.

7. PLOS authors have the option to publish the peer review history of their article (what does this mean?). If published, this will include your full peer review and any attached files.

Reviewer #2: **Yes: **Robin L Peterson

---

## [Author Response · Author response to Decision Letter 1]

27 Jul 2021

We thank the reviewer for their useful feedback on our manuscript; the resulting changes have clearly improved our paper and the clarity of its contributions, and we now feel it is ready for publication in PLOS ONE. Please see the table attached where we detail how we have explicitly addressed each of the reviewer comments.

---

## [Editor Report · Decision Letter 2]

30 Jul 2021

The assignment and distribution of the dyslexia label: Using the UK Millennium Cohort Study to investigate the socio-demographic predictors of the dyslexia label in England and Wales

PONE-D-21-02074R2

Dear Dr. Knight,

We’re pleased to inform you that your manuscript has been judged scientifically suitable for publication and will be formally accepted for publication once it meets all outstanding technical requirements.

Kind regards,

Enrico Toffalini, Ph.D

Academic Editor

PLOS ONE
---

## [Editor Report · Acceptance letter]

5 Aug 2021

PONE-D-21-02074R2 

The assignment and distribution of the dyslexia label: Using the UK Millennium Cohort Study to investigate the socio-demographic predictors of the dyslexia label in England and Wales 

Dear Dr. Knight:

I'm pleased to inform you that your manuscript has been deemed suitable for publication in PLOS ONE. Congratulations! Your manuscript is now with our production department. 

Kind regards, 

on behalf of

Dr. Enrico Toffalini 

Academic Editor

PLOS ONE